# Symbiotic Fungus *Serendipita indica* as a Natural Bioenhancer Against Cadmium Toxicity in Chinese Cabbage

**DOI:** 10.3390/plants14172773

**Published:** 2025-09-04

**Authors:** Akram Rahbari, Behrooz Esmaielpour, Rasoul Azarmi, Hamideh Fatemi, Hassan Maleki Lajayer, Sima Panahirad, Gholamreza Gohari, Federico Vita

**Affiliations:** 1Department of Horticulture, Faculty of Agriculture and Natural Resources, University of Mohaghegh Ardabili, Ardabil 5619911367, Iran; rahbari.a94@gmail.com (A.R.); r_azarmi@uma.ac.ir (R.A.); 2Department of Horticulture, Ferdowsi University of Mashhad, Mashhad 9177948974, Iran; ha.fatemi@yahoo.com; 3Faculty of Agriculture (Meshkin Shahr Campus), University of Mohaghegh Ardabili, Ardabil 5619911367, Iran; malekih135@gmail.com; 4Department of Landscape Engineering, Faculty of Agriculture, University of Tabriz, Tabriz 5166616471, Iran; s.panahirad@tabrizu.ac.ir; 5Department of Horticultural Sciences, Faculty of Agriculture, University of Maragheh, Maragheh 551877684, Iran; gohari.gh@maragheh.ac.ir; 6Department of Biosciences, Biotechnology and Environment, University of Bari “Aldo Moro”, 70121 Bari, Italy

**Keywords:** endophytic fungus, heavy metal, secondary metabolites, osmolytes, antioxidant compounds

## Abstract

Heavy metal toxicity, particularly cadmium (Cd), poses a growing threat to agriculture and human health due to its persistence and high solubility, which facilitates its entry into the food chain. Among the strategies proposed to reduce Cd toxicity in plants and the environment, the use of beneficial microorganisms, such as endophytic fungi, has gained attention due to its effectiveness and eco-friendliness. This study investigates the potential of the root-colonizing fungus *Serendipita indica* (formerly *Piriformospora indica*) to mitigate cadmium (Cd) stress in Chinese cabbage (*Brassica rapa* L. subsp. *Pekinensis*) grown hydroponically under varying Cd concentrations (0, 1, 3, and 4 mM). Several parameters were assessed, including morphological traits, physiological and biochemical responses, and changes in leaf composition. Exposure to Cd significantly reduced plant growth, increased membrane electrolyte leakage, and decreased relative water content and root colonization, while enhancing antioxidant enzyme activities and the accumulation of phenolics, flavonoids, proline, glycine betaine, and carbohydrates. Notably, plants treated with *S. indica* showed improved tolerance to Cd stress, indicating the potential of the fungus. These findings suggest that *S. indica* can enhance plant resilience in Cd-contaminated environments and may offer a promising biological strategy for sustainable crop production under heavy metal stress.

## 1. Introduction

In recent decades, industrialization and pollution caused by agricultural sources, such as chemical fertilizers, have increased the production of urban and industrial waste, leading to an increase in the level of heavy metals in the environment [1]. Accumulation of heavy metals (e.g., cadmium (Cd), nickel (Ni), lead (Pb), copper (Cu), zinc (Zn)), as the most dangerous mineral pollutants due to non-biodegradability and long-term persistence in the soil and in agricultural products, leads to health problems in living organisms, especially humans [2].

The Association for Toxic Substances and Disease Registry (ATSDR) reported that Cd was ranked seventh among 20 hazardous substances [3]. Plants are the main route of Cd entering the human food chain, and Cd’s high concentrations in plants lead to disturbances in their physiological, morphological, and biochemical functions [4]. The Cd accumulation in roots hinders root growth and nutrient absorption, resulting in alterations in enzymatic functions and plant organ structures [5]. Likewise, an increase in Cd amount leads to a decrease in chlorophyll content, photosynthesis rate, and the relative content of leaf water, as well as physiological processes such as the reduction of intracellular spaces, the number of chloroplasts, and the number and size of xylem vessels in the leaves of stressed plants [6].

Various chemical, physical, and biological approaches have been employed to mitigate the toxicity of heavy metals in contaminated environments [7]. Among these methods, bioremediation, including applying bacteria, fungi, and algae, is more cost-effective with lower environmental harm and can serve as a replacement for methods with harmful effects to the soil, plants, and humans [8].

Recently, the endophytic fungus *Serendipita indica* (formerly known as *Piriformospora indica*), known for its pear-shaped chlamydospores, represents one of the microorganisms of interest in bioremediation [9]. The fungus belongs to the Sebacinales order and plays an important role in improving yield, growth, and stress tolerance in plants. *S. indica* was isolated from the rhizosphere of desert plants in India in 1998 [10]. Through numerous studies, it has been found that *S. indica* application has positive effects, including promoting water absorption, increasing stress resistance, and enhancing yield in inoculated plants [11,12].

*S. indica*, by causing changes in the hormones of the host plant, leads to the activation of the antioxidant system and increases tolerance to abiotic stresses in inoculated plants, improving plant growth [13]. The fungus reduces Cd transfer from roots to plant tissues, indicating its capacity as a suitable treatment in Cd-contaminated soils [14]. It also leads to an increase in the absorption of nutrients in plants by enhancing the length and quality of their roots, thereby improving their nutritional status [15].

It has been reported that inoculation of rice, Chinese cabbage, and maize plants with *S. indica* fungus increased root growth and plant biomass [16]. In areas where phosphorus is deficient, inoculation of plants with *S. indica* leads to increased phosphorus uptake in the inoculated plants [17]. Increased phosphorus uptake in plants increases nitrogen compounds, improves photosynthesis, reduces malondialdehyde content, and maintains the relative water content of leaves [18]. *S. indica* protects plants under biotic and abiotic stress conditions by increasing the production of proline, sucrose, glycine, and betaine, as well as antioxidant enzymes such as catalase and superoxide dismutase [19]. In this view, one of the important applications of *S. indica* is to increase plant resistance to abiotic stresses such as heavy element toxicity [20].

It has been reported that the chelation of Cd ions by *S. indica* or the absorption of Cd in the cell wall of the fungus leads to a decrease in the Cd accumulation in the plant root, which then prevents Cd transfer from the root to the plant’s shoots [21].

Nowadays, vegetables have an essential role in the human diet because they contain nutrients, act as a main source of minerals, and reduce the risk of some diseases [22]. Chinese cabbage (*Brassica rapa* L. subsp. *Pekinensis*) is an important leafy vegetable of the genus *Brassica* with high economic and agricultural importance thanks to its high nutritional value, fast growth, and low production cost, which is widely cultivated worldwide. Brassica plants are widely used as fresh food, seasonings, and oilseeds. Among the Brassica plants, Chinese cabbage is one of the most important and widely consumed vegetables in Asia and is now widely cultivated worldwide [23]. Chinese cabbage is rich in several nutrients, such as vitamins, carotenoids, and glucosinolates (which have anti-aging, anti-cancer, and antioxidant effects) [24].

Significantly, the growth and quality of this vegetable are reduced under cadmium stress. Therefore, cadmium accumulation in this vegetable is a serious threat to food safety and, consequently, human health [25].

In the current research, the effect of *S. indica* on reducing the adverse effects of Cd stress on Chinese cabbage was investigated under hydroponic cultivation using a set of morphological, physiological, and biochemical descriptors of Chinese cabbage under Cd stress or not under Cd stress for describing plant physiological responses.

## 2. Results

### 2.1. Morphological Traits

Cd stress and inoculation with *S. indica* significantly affected the rate of shoot growth, such that cadmium reduced plant growth and inoculation with fungus was able to reduce cadmium toxicity in the plant (Figure 1). Cd stress and inoculation with *S. indica* significantly affected plant height and fresh (FW) and dry (DW) weights of shoots (Figure 2). The factor interaction Cd stress × *S. indica* is significant in plant height and fresh weight traits. Chinese cabbage plant height was significantly decreased by Cd stress (Figure 2A). Shoot fresh weight was additionally decreased by Cd application (the higher Cd levels, the lower plant height; 34% decrease under 4 mM Cd), and inoculation with *S. indica* reduced the effects of Cd stress by increasing fresh weight under non-stress and 1 mM Cd-stress conditions, while it had no effect under 3 and 4 mM Cd conditions (Figure 2B). The concentration of 3 and 4 mM Cd stress significantly reduced the shoots’ dry weight (34% decrease under 4 mM Cd) (Figure 2C). Overall, *S. indica* inoculation affects the dry weight by increasing the plant shoot dry weight by 14% compared to the control (Figure 2D).

### 2.2. Electrolyte Leakage (EL) and Relative Water Content (RWC)

According to the results, EL values were affected by the Cd stress, inoculation, and their interaction (Figure 3). The increase in Cd concentration resulted in a corresponding increase in the EL amount, with higher Cd levels associated with higher EL (a 53% increase in EL at 4 mM Cd) in both control and *S. indica* conditions. The inoculation led to a decrease in the EL values in all stress conditions (0, 1, 3, and 4 mM Cd) in *S. indica* compared to the control, thus indicating that S. fungal inoculation mitigates the stress effects (Figure 3A). RWC was affected by Cd stress levels and slightly by inoculation; however, their interaction had no significant effect in this regard (Figure 3B). The results showed that the RWC of Chinese cabbage leaves decreased significantly with an increase in Cd concentrations (a 38% reduction at 4 mM Cd) (Figure 3B). Also, plants inoculated with the fungus had a 20% higher RWC than non-inoculated (control) plants (Figure 3B).

### 2.3. Proline and Carbohydrate Contents

Cd stress impacts plant physiology by affecting the compatible osmololyte (proline, glycine betaine) and carbohydrate contents, despite factor interaction being statistically significant for glycine betaine data. Cd stress at all applied levels significantly enhanced proline content (84% increase at 4 mM Cd) (Figure 4A), with a general increase in inoculated plants compared to non-inoculated (control). Glycine betaine (GB) data (Figure 4B) indicate that stress severity impacts the GB content; an increase in Cd levels considerably enhanced GB content compared to the control (35% increase at 4 mM Cd). In this view, *S. indica* inoculation caused a significant increase in GB content under non-stress conditions while causing no significant change under Cd stress conditions (Figure 4B). Moving to total soluble carbohydrates (Figure 4C), data indicate that inoculation with *S. indica* caused an increase in the amount of carbohydrates. Notably, the most impactful stress conditions (3 and 4 mM Cd) showed a more pronounced difference between inoculated and non-inoculated (control) samples. Osmolyte data were then analyzed through linear correlation analyses (Figure 4D) to assess whether the data for proline and glycine betaine were correlated in the analyzed samples. Results indicate that a certain degree of correlation among osmolyte data occurs (R^2^ = 0.7486).

### 2.4. Enzyme Activities and Total Metabolites

Catalase and Superoxide dismutase enzyme activities were affected by Cd stress and *S. indica* inoculation; however, their interaction was not significant in this regard. In general, Cd stress levels meaningfully enhanced Catalase and Superoxide dismutase activities (165% and 65% at 4 mM Cd, respectively) (Figure 5A,B), thus indicating that the inoculation with *S. indica* caused a remarkable increase in Catalase and Superoxide dismutase activities as compared to the non-inoculated (control) plants.

Two-way ANOVA analyses revealed that the effects of Cd stress and inoculation with *S. indica* were significant on total phenolic and flavonoid indices (Figure 5C,D) for both sets of variables (Cd concentration and mycorrhization), although their interaction was not significant. A substantial increase in both total phenols and flavonoids was observed, which is directly proportional to the severity of stress, reaching a 25% and 43% increase at 4 mM Cd, respectively, compared to non-stress conditions. Additionally, *S. indica* fungus inoculation resulted in a general increase in total phenols and flavonoid content (12% and 4% compared to non-inoculated (control) plants, respectively).

### 2.5. Elements

#### 2.5.1. Cd

Cd stress conditions, *S. indica* inoculation, and their interaction had a significant effect on the Cd content of roots and shoots (Figure 6A,B). An increase in Cd levels resulted in a remarkable rise in Cd content in both shoot and root parts, reaching the highest values at 4 mM Cd. Also, significant differences occurred for all stress conditions (1, 3, and 4 mM Cd) between inoculated and non-inoculated (control) plants, with generally higher Cd values reported in both roots and shoots, except for the 4 mM Cd root data, where inoculated and non-inoculated (control) plants had comparable values.

#### 2.5.2. N

Nitrogen accumulation data showed distinct trends for roots (Figure 6C) and shoots (Figure 6D). The interaction effect of Cd stress and fungal inoculation on the nitrogen content for root data is significant, whereas, in contrast, the nitrogen content in the shoots was affected only by Cd stress and fungal inoculation, with no significant interaction effect between these two treatments. With increasing Cd toxicity, a significant decrease in N content was observed in the roots and shoots, even if the decrease was more pronounced in shoots (Figure 6D). Additionally, fungal inoculation increased the nitrogen content in the shoots under all stress conditions for both tissues. The highest nitrogen content in the roots was observed in plants inoculated under non-stress conditions and 3 mM Cd (Figure 6C).

#### 2.5.3. P

Results related to P determination showed that both variables, as well as their interaction, were statistically significant in both tissues (Figure 6E,F). P content significantly decreased in Cd-treated plants (increased Cd levels reduced P content) proportionally with stress severity. Additionally, inoculation increased the P content of both the roots and shoots under both non-stress and Cd-stress conditions. The highest and lowest P content values were recorded in *S. indica* inoculated and 4 mM Cd-stressed plants, in that order (Figure 6E,F).

#### 2.5.4. K

The K content of roots and shoots was affected by Cd stress (Figure 6G,H), as well as by fungus inoculation and their interaction. An increase in Cd levels resulted in a decrease in K content in the roots and shoots, and inoculation with the fungus enhanced the content under both non-stress and Cd-stress conditions. The highest K content (roots and shoots) was achieved in inoculated plants under non-stress conditions (Figure 6G,H).

#### 2.5.5. Ca

Cd stress, *S. indica* inoculation, and their interaction affected the Ca content of shoots, while only Cd stress and *S. indica* inoculation altered the Ca content of the roots (Figure 7A,B). Ca content was negatively affected by Cd stress conditions (the higher Cd levels, the lower Ca content of the shoots and roots), with a more pronounced decline observed in the shoot at 4 mM Cd (Figure 7B). Inoculation with the fungus led to an increase in the Ca content of the shoot, regardless of Cd stress conditions. A similar trend was also observed for root and shoot data (Figure 7A,B), where Ca content was enhanced by *S. indica* inoculation in all stress conditions, with the highest value recorded in inoculated plants under non-stress conditions.

#### 2.5.6. Fe

Both variable sets (Cd concentration and mycorrhization) yielded significant results in both tissues, despite their interaction not being significant (Figure 7C,D). An increase in Cd levels resulted in a significant reduction in Fe content in both roots and shoots, with the lowest amount recorded at 4 mM Cd. Inoculation with *S. indica* positively affected the Fe amount of both roots and shoots, with significant increases with respect to non-inoculated (control) counterparts. The root and shoot Fe amount in *S. indica* inoculated plants increased compared to non-inoculated (control) plants (Figure 7C,D).

### 2.6. Microbial Population

The effect of Cd stress, inoculation with *S. indica*, and their interaction was significant for microbial population data (Figure 8A). Cd stress leads to a remarkable decrease in the population. Also, inoculation with *S. indica* significantly enhanced the population under non-stress and Cd-stress conditions. According to the results, the highest microbial population was obtained in inoculated Chinese cabbage plants grown under non-stress conditions (Figure 8A). Moving to data related to symbiosis percentage (Figure 8B), Cd stress, inoculation with *S. indica* fungus, and their interaction significantly affected this parameter. The stress impact is noticeable, as it markedly affects the fungal symbiosis at 3 and 4 mM. Nevertheless, symbiosis percentages increased in inoculated plants as a consequence of *S. indica* application, with the highest symbiosis percentage reported in inoculated plants under non-stress conditions (Figure 8B).

### 2.7. Scanning Electron Microscope (SEM) Observations

Scanning electron microscopic (SEM) observations of Chinese cabbage leaves treated with Cd and inoculated with *S. indica* showed significant changes (Figure 9 and Figure 10). The SEM micrographs from the epidermis of the Cd-treated plants indicated that leaf stomatal closure occurred, especially at higher stress levels (Figure 10H). The decrease in the diameter of the stomata in Cd-treated leaves was additionally observed, particularly at 3 and 4 mM concentrations compared to non-stress ones (Figure 10G,H). Moreover, the SEM micrographs of non-stressed leaves have no visible abnormalities in both inoculated and non-inoculated (control) plants (Figure 9A,E). Furthermore, plants inoculated with *S. indica* fungus under Cd stress conditions showed resistance to stomatal closure and reduction in stomatal diameter compared to non-inoculated (control) plants, with stomata characterized by a wider opening. These results support the hypothesis that the fungus mitigates the stress effect by inducing a condition similar to non-stress, thereby leading to improved stomatal activity under Cd stress conditions (Figure 9A).

## 3. Discussion

According to the results obtained in the present study, it was shown that Cd at different concentrations caused disruption in the growth performance and physiological and biochemical characteristics of Chinese cabbage. In contrast, inoculation with *S. indica* fungus increased these parameters and resulted in better plant growth. Since Cd is a toxic element, its destructive effects on the morphological and physiological structure of plants are expected to cause a decrease in growth and biomass [26]. Studies conducted in spinach (*Spinacia oleracea*) [27], cucumber (*Cucumis sativus*) [28], and tomato (*Solanum lycopersicum*) [29] confirmed the Cd adverse effect on growth parameters.

It has been determined that *S. indica* activates different mechanisms to enhance tolerance to Cd stress, which could improve the growth rate and stimulate plant growth. As a result, plants can tolerate the stress [30]. Since the plant root vacuole is considered the main Cd accumulation site, *S. indica* fungus could increase Cd symbiosis in the vacuole of inoculated plants and prevent its transfer to other parts of the plant [31]. Likewise, inoculation of plants such as rice, wheat, sunflower, corn, and barley with *S. indica* fungus positively affected plant growth under Cd stress [26].

Rucińska-Sobkowiak (2016) reported that lettuce and bean plant growth decreased under Cd stress due to Cd-induced water shortage that caused reduced leaf RWC, confirming our results regarding the decrease in the RWC of Chinese cabbage leaves under Cd stress [32].

The present study showed that Cd stress increased EL contents. Similar results were reported by Hashem et al. [33] on *Solanum lycopersicum* and Sheikhalipour et al. [34] on *Stevia rebaudiana Bertoni*. By increasing ROS content, Cd leads to the intensification of several stress markers, ultimately increasing EL [35]. Inoculation of Cd-stressed plants with *S. indica* prevents membrane lipid peroxidation and plant cell death by neutralizing ROS and increases membrane stability [36]. Accumulating compatible osmolytes, such as proline, glycine betaine, and free sugars, is the most common strategy of plants to tolerate stressful conditions [37].

It has been reported that various osmolytes (proline, carbohydrates, and glycine betaine) lead to improved plant performance by maintaining water balance and improving nutrient uptake, maintaining membrane structures, and eliminating ROS [38]. In the present study, we observed that sugar accumulation occurred in fungal-inoculated plants under Cd stress. This was consistent with the results of Islam et al. [39]. Several reports of sugar accumulation during heavy metal stress include Cd [40].

The high total sugar content in Cd-treated seedlings may be due to high alpha-amylase activity or poor sugar utilization during early growth stages in stressed plants. Microorganisms lead to the accumulation of soluble sugar in the inoculated plants, and photosynthesis is reduced, thereby maintaining homeostasis [41]. Additionally, they play a key role in scavenging radicals [42].

In the present study, proline levels increased under Cd stress and fungal inoculation, in line with the previous report in tomato plants under Cd stress [38]. Increased proline content, as an osmotic regulator, is the most common sign of abiotic stress tolerance in plants, which is essential for reducing stress toxicity [34], as observed in the current study, especially at 4 mM Cd concentration. Proline could act as a metal chelator that accumulates Cd ions in the cytosol and then transfers them into the vacuole. Increased proline content in Cd-stressed plants increased the resistance through four pathways: eliminating ROS, improving the activity of plant enzymes, chelating Cd, and maintaining water balance in plant cells and tissues [11], all of which could explain our results.

Inoculation with *S. indica* enhanced the proline content in Cd-stressed plants [43], which is in line with the current study.

Glycine betaine (GB), an N,N,N-trimethylglycine, is a key osmoprotectant [44]. GB reduces oxidative stress in plants by increasing the production of enzymatic and non-enzymatic antioxidants [45].

The highest GB accumulation in Chinese cabbage was observed in the 4 mM cadmium treatment. Many studies are showing the positive effects of GB on reducing metal stress in different plant species [46]. The Cd strongly binds with the functional group (-SH refers to thiol or sulfhydryl group) and the competition for replacement with cations such as calcium, magnesium, and zinc, which are essential for enzymatic activation, cause inhibition of the activity of the main enzymes in plants [47].

Plants use enzymatic (Catalase (CAT), Glutathione reductase (GR), Peroxidases (POX), Superoxide dismutase (SOD), and Ascorbate peroxidase (APX)) and non-enzymatic (proline, soluble sugars, phenolic compounds, glutathione, and carotenoids) antioxidant systems to reduce damage caused by ROS [48]. Increasing the levels of antioxidant enzymes leads to eliminating excess ROS generated by heavy metal stress in cells, which helps reduce ROS-induced damage. Many studies have shown that increasing various antioxidant enzymes can increase resistance in plants under stress [49]. It has been reported that tolerance to Cd stress occurs when ROS are removed or detoxified by antioxidant enzymes such as Catalase and Superoxide dismutase that can absorb free radicals as an essential defense system in plants [50]. Superoxide dismutase is an important antioxidant enzyme and the first and most effective line of defense against ROS. Superoxide dismutase protects plant cell components against abiotic and biotic stress [51].

In the present study, the activities of Superoxide dismutase and Catalase enzymes increased with *S. indica* inoculation, especially at the 4 mM Cd level. Similar results have been reported in different plant species, such as sunflower (*Helianthus annuus*) [52], bean (*Phaseolus vulgaris* L), and tomato (*Solanum lycopersicum*) [53]. The increase would help the plant mitigate stress-induced damage. Furthermore, this conclusion is consistent with the results obtained from the use of *S. indica* in reducing oxidative damage and increasing growth in cabbage [54], tomato [55], and wheat [56] under stress.

Non-enzymatic antioxidants (e.g., phenolic compounds, flavonoids) could absorb free radicals as an essential defense system in plants [50]. Phenols and flavonoids act as non-enzymatic antioxidants and reduce oxidative stress in plants exposed to stress [57]. In the present study, Cd stress increased the amount of phenols and flavonoids in Chinese cabbage, consistent with the findings of Pawlak-Sprada et al. [58], which states that phenols and flavonoids protect plants under heavy metal stress due to their metal chelating and antioxidant properties. Several reports show that microbes regulate phenol metabolism by producing shikimic acid, a critical metabolite in phenol biosynthesis [59].

In the present study, the inoculation of Chinese cabbage with *S. indica* fungus caused a reduction in the amount of Cd accumulation and absorption by roots and shoots and then led to an improvement in plant growth, which was consistent with the results obtained by [60].

Phosphorus, potassium, and nitrogen are the most important mineral nutrients for plants, which are required for protein activation and amino acid formation [61]. Depletion of potassium, nitrogen, and phosphorus due to stress from heavy metals causes necrosis and chlorosis, reduced plant growth, and loss of turgor [62]. Cd led to a decrease in the uptake of nitrogen elements by Chinese cabbage plants because Cd competes with the uptake and transport of minerals [63], as observed in the current study. Numerous reports indicate that bacteria fix nitrogen by activating the nif gene and other effector genes, which can also lead to increased plant yield, improved plant growth, enhanced soil properties, and increased soil nitrogen retention [61]. Additionally, *S. indica* enhances plant performance by increasing nitrogen absorption through the production of NO signals and the expression of nitrate reductase [64].

Heavy metals can also disrupt the absorption of phosphate by plants and inhibit growth [51]. *S. indica* increased the uptake and transport of phosphate by plant roots and improved the plant’s access to this nutrient by increasing the expression of phosphate transporter genes [10]. As a result, the increased P absorption in inoculated plants preserved leaf RWC and increased plant performance [18].

In this study, the amount of potassium in the shoots and roots of Chinese cabbage decreased. The decrease in nutrient absorption under heavy metal stress could be due to the decrease in root branching and length and an increase in root thickness [65]. Inoculation of Chinese cabbage plants with *S. indica* fungus under Cd stress resulted in increased potassium uptake in the plant. It has been reported that Plant growth-promoting bacteria (PGPB) increase plant growth under abiotic stress by producing ACC-deaminase, an enzyme that increases the uptake of essential nutrients such as K [66].

Due to its chemical behavior and charge similarity, Cd can substitute for the essential calcium ion and be transported to the above-ground parts of the plant [44]. Cd also penetrates the cytosol through calcium channels in the plasmalemma, leading to changes in cell–water relations in stressed plants [67]. It has been reported that *S. indica*, with high uptake potential, helped decrease cadmium toxicity and increase the concentration of Ca ions under Cd stress. This may be due to the improvement of soil organic matter and increased nutrient availability by *S. indica* in the Cd-contaminated growing medium [68].

Cd has been shown to compete with iron in the transport of iron across membranes [69]. Several studies have confirmed the reduction of mineral elements, including iron, in plants under Cd stress [70]. Researchers have stated that microorganisms, such as *S. indica*, which have the ability to produce siderophores, improve plant growth under Cd stress compared to organisms that do not produce such compounds [11].

In summary, inoculation with *S. indica* increased Cd tolerance in Chinese cabbage. Chinese cabbage plants treated with Cd showed reduced plant growth, essential nutrient uptake, and RWC. In contrast, increases in carbohydrate, proline, and oxidative stress indices (MDA and EL) were also observed. The results showed that application of *S. indica* increased the growth, nutrient uptake, RWC, enzymatic and non-enzymatic antioxidant activities such as Superoxide dismutase, Catalase, phenolics, flavonoids, and glycine betaine, and reduced MDA and EL levels under Cd toxicity. Accordingly, it was observed that inoculation with *S. indica* has the potential to reduce Cd stress effects by improving the antioxidant defense system and modulating ROS inhibition.

## 4. Materials and Methods

### 4.1. Plant Material and Growth Conditions

This research was carried out in the research greenhouse of the University of *Mohaghegh Ardabili*, Iran, with an average relative humidity of 60%, in the fall of 2022 and winter of 2023, in three replications as a factorial in a completely randomized design under hydroponic conditions. In the present study, the effect of cadmium metal at four levels (0, 1, 3, and 4 mM) and the effect of fungus (no inoculation with fungus and inoculation with fungus) on morphological, physiological, and biochemical indices of Chinese cabbage plants were investigated.

In this study, first, fungus screening was performed. In order to screen fungus for cadmium tolerance, they were cultured on plates containing Hill and Kuffer medium at concentrations of 0, 0.25, 0.5, 1, 1.5, 2, 2.5, 3, 3.5, 4, 4.5, and 5 mM and incubated for 72 h at 28 ± 2 °C; then, the growth or lack of growth of the fungus was examined. To carry out this study, Chinese cabbage seeds (Summer Queen F1) produced by Asia Seed Company of South Korea were purchased. *Serendipita indica* endophyte was obtained from the Soil Science Department, University of Mazandaran [68,71]. To prepare *S. indica* mycelium, 5 mm active fungal discs were isolated from Hill and Kaefer solid medium and transferred to 100 mL of Hill and Kaefer liquid medium in 250 mL Erlenmeyer flasks and then placed in a shaker incubator (110 rpm) at 28 ± 2 °C for 7–10 h. Next, the mycelium was separated from the liquid suspension by filtering through Whatman paper No. 1 and washed several times with sterile distilled water; excess water was discarded.

### 4.2. Seedbed Preparation

For this experiment, Chinese cabbage seeds were soaked in 1.5% sodium hypochlorite for 15 min and then washed four times with deionized water. The obtained fungal endophyte mycelium was thoroughly mixed with sterile culture medium, containing cocopeat and perlite (1:1) at a concentration of 1% (*w*/*v*). A combination of cocopeat and perlite is often used as a growing medium in container gardening and hydroponic systems. Cocopeat provides water retention and aeration, and perlite improves drainage and aeration, as well as preventing waterlogging, thus providing a balanced medium for plant growth [72]. Then, sterilized Chinese cabbage seeds were placed in each hole of the planting tray. After germination and seedling transfer, half-strength Hoagland’s solution (500 mL/day) was used in all treatments [73]. The hydroponic solution contained the macronutrients (mg/L) nitrate (N) 147, potassium (K) 163.8, calcium (Ca) 140, sulfur (S) 43.4, magnesium (Mg) 33.6, and phosphorus (P) 21.7, and the micronutrients (mg/L) iron (Fe) 3.5, boron (B) 0.35, manganese (Mn) 0.35, copper (Cu) 0.01, zinc (Zn) 0.035, and molybdenum (Mo) 0.007. The pH of the hydroponic solution was measured at 6.3; due to the use of solid growing medium, aeration was not required. After plant establishment (one week after transplanting seedlings to pots), cadmium chloride hydrate with a purity of more than 80% was added to the nutrient solution at concentrations of 0, 1, 3, and 4 mM (obtained after screening), and the plants were irrigated with Hoagland’s solution until harvest. Control plants were irrigated with Hoagland solution without the fungus inoculation and without Cd contamination in the same manner until harvest. Chinese cabbage is harvested when the cabbage head is large and firm enough. In the present study, the plant was harvested 55 days after the seedlings were transferred to the pots until the head was closed. All pots were maintained under the same greenhouse environmental conditions with an average temperature of 25 °C and humidity of 40%.

### 4.3. Measurement of Morphological Traits

Chinese cabbages were harvested at the end of the experiment. Sample tissues (roots and shoots) were separated and transferred from the greenhouse to the laboratory. Biometric data (shoot length) was measured; then, leaves and roots were placed in a paper bag and transferred to an oven at 70 °C for 72 h. Dry and wet weight data were collected using a digital scale with an accuracy of 0.001 g.

### 4.4. Relative Water Content (RWC)

A total of 0.5 g of leaf sample from the youngest developed leaves of each plant was weighed (FW) and then floated in distilled water for 24 h. Next, the saturated weight of the leaf was measured (TW). Then, the leaves were placed in an oven (70 °C, 24 h), and their dry weights were measured. Finally, RWC was obtained based on the following formula [74]:$$\%RWC=\frac{FW-DW}{TW-DW}$$

### 4.5. Electrolyte Leakage (EL)

Electrolyte leakage was evaluated using spectrophotometry, which is a relatively simple and sensitive method and has the ability to measure multiple ions simultaneously. In order to measure the relative permeability, 0.5 cm diameter leaf disks were prepared from fully developed leaves and washed three times with deionized water. The samples were placed in a closed container containing 10 mL of deionized water and shaken on a shaker for 24 h at 25 °C. After that, the initial electrical conductivity (EC1) was read by a portable EC meter (MILWAUKEE MW170 MAX). Then, the samples were placed in the water bath (95 °C, 20 min), and EC was read again (EC2). The following formula was used for calculation [75]:EL (%) = (EC1/EC2) × 100

### 4.6. Soluble Carbohydrates

The amount of soluble carbohydrates was calculated using the method of Irigoyen et al. [76]. In this method, 0.1 g of fresh leaves of the plant were finely ground in a porcelain mortar with 5 mL of ethanol (80%) and placed in a hot water bath (70 °C) for 10 min. The extract obtained was centrifuged for 10 min at 11,180× *g*. The upper clear part was separated and used to measure the amount of soluble sugars. Then, 3 mL of anthrone solution was added to 100 μL of the extract, and the mixture was placed in a water bath at 100 °C for 20 min. The amount of soluble sugars was then measured using a spectrophotometer (Hitachi U-2910, Tokyo, Japan) at a wavelength of 625 nm. Finally, the soluble carbohydrate content was quantified using a glucose-based reference standard curve.

### 4.7. Proline Content

To measure proline content, 0.5 g of fresh leaf tissue was ground with 10 mL of 3% sulfosalicylic acid in a porcelain mortar for 3 min. A total of 2 mL of the extract, 2 mL of ninhydrin, 2 mL of pure glacial acetic acid, and toluene were poured into a test tube. The tubes were vortexed for 15 to 20 s. After forming two separate phases, the upper colored phase was carefully separated, and the absorbance was measured using a spectrophotometer (Hitachi U-2910, Tokyo, Japan) at a wavelength of 520 nm. Proline content was determined using its standard curve [77].

### 4.8. Glycine Betaine

The dried leaves (0.5 g) were mixed with 20 mL of distilled water and placed on a shaker at 25 °C for 24 h to obtain the plant extract. The plant extract (1 mL) was mixed with sulfuric acid (2 N, 1 mL) and placed in an ice water bath; then, potassium iodide (0.2 mL) was added to the reaction mixture. Then, it was centrifuged at 11,180× *g* for 15 min at 0 °C. The supernatant was carefully collected. The periodate crystals were dissolved in 9.0 mL of 1,2-dichloroethane and mixed. After 2 h, the absorbance at 365 nm was measured using a spectrophotometer (Jenway6705, Spain). Glycine betaine was also used to draw a standard curve [78].

### 4.9. Antioxidant Enzyme Activities

#### 4.9.1. Protein Extraction

Protein extraction was performed using 0.2 g of fresh leaf tissues homogenized with 1 mL phosphate buffer (600 mM, pH = 7.5) in a mortar. Then, samples were centrifuged (150,975× *g*, 4 °C, 20 min), and the supernatant was kept at −80 °C for further investigation. The reaction mixture contained 100 μL of enzyme extract and 900 μL of Bradford reagent, and absorbance was measured at 595 nm [79].

#### 4.9.2. Catalase (CAT) Enzyme Activity

Protein extract (5 µL) was mixed with hydrogen peroxide (H_2_O_2_, 10 mM) and potassium phosphate buffer (495 µL) containing 0.1 mM EDTA. Enzyme activity was recorded as absorbance changes over time at a wavelength of 240 nm using a spectrophotometer (Jenway 6705) for one minute. Finally, the enzyme activity was calculated using the extinction coefficient of 39.4 mM^−1^ cm^−1^ and the Beer–Lambert law formula (µmoL min^−1^ g^−1^ FW) [80].

#### 4.9.3. Superoxide Dismutase (SOD) Enzyme Activity

In order to measure Superoxide dismutase activity, a 3 mL reaction mixture containing 50 mM potassium phosphate buffer with 13 mM methionine, 75 μM nitrobuterazolium, 20 μM riboflavin, 0.1 μM EDTA, and 100 μL enzyme extract was exposed to 5000 lux light for 15 min. Then, using the spectrophotometer, the absorbance of the samples was read at a wavelength of 560 nm. Enzyme activity was calculated using the quenching coefficient of 100 mM^−1^ cm^−1^ (µmoL min^−1^ g^−1^ FW) [81].

### 4.10. Total Phenol and Flavonoid Contents

To prepare the ethanolic extract of Chinese cabbage, 100 g of powdered dry Chinese cabbage leaf samples was used according to the protocol of Vega-Gálvez et al. [82], and a 10% (*w*/*v*) methanolic extract was prepared by soaking and subsequently centrifuged. After that, first, a 750 μL Folin–Ciocalteu indicator (diluted in a 1:10 ratio) was added to 100 μL of the extract. Then, 4 mL of 7.5% sodium carbonate was added after 5 min. The mixture was incubated at room temperature (dark condition) for 90 min. Last, the absorbance was read at a wavelength of 750 nm by the spectrophotometer.

For flavonoid content determination, 0.2 mL of the extract, 90% ethanol, 4.5 mL of aluminum chloride 2%, and 0.2 mL of acetic acid 33% were mixed and then placed at room temperature in dark conditions for 30 min. Finally, the absorbance of the samples was read at 414 nm [82].

### 4.11. Analysis of Ion Contents in the Biomass

The prepared leaf samples were dried in an oven for 24 h at 65 °C and then ground to a uniform powder. Then, one gram of the powdered sample was poured into a porcelain crucible and placed in a furnace at 550 °C for 5 h. Samples were then cooled, and the ashes were mixed with 10 mL of 2 N hydrochloric acid. The mixture was added and placed on a heater with gentle heat to facilitate dissolution. The solution was filtered using filter paper (Whatman 42) and poured into a 50 mL flask. Finally, the prepared extract was made up to a volume of 50 mL. The Cd amount present in the extracts prepared from the leaf and root was read separately by an atomic absorption device (Perkin Elmer 400AA model, Perkin-Elmer, Waltham, MA, USA).

Nitrogen (N) content was measured according to the [83] method. A total of 0.1 g of the dried sample was added to the digestion tube; then, 10 mL of concentrated sulfuric acid and salicylic acid were added to the tube and placed in the Kjeldahl digestion apparatus for 2.5 h. After digestion, the sample was distilled with an automatic Kjeldahl apparatus for N separation. Then, it was titrated with 0.01 normal hydrochloric acid, and the N content of each sample was determined using the following formula:Mg N/Gms= (V − V0) mL × T(Meq/mL) × 14(mg/Meq)N × 1/0.1 gMSV = Volume of H_2_SO_4_ added to the sampleV0 = Volume of H_2_SO_4_ added to the blankT = H_2_SO_4_ normalityMS = Dry matter (sample)

To measure P content, 10 mL of hydrochloric acid (2 M) was added to the plant samples, transferred to the balloon, and brought to 1000 mL with distilled water. After calibrating, the absorbance of the samples was recorded at a wavelength of 420 nm using the spectrophotometer. The P amount was calculated through the following formula:P = (a − b) × (v/s) × mcf

a, the P amount in the extracted sample; b, the P amount in the blank; v, extraction solution; s, the weight of plant sample; mcf, a humidity correction factor [84].

First, the leaves and roots of Chinese cabbage were ashed in an oven at 550 ± 25 °C. To the obtained ash, 10 mL of concentrated hydrochloric acid was added, and the volume was made up to 100 mL. Potassium concentration was measured using a flame photometer [85], calcium was measured using the method described by Saini et al. [86], and iron was measured using an atomic absorption spectrometer [87]. Cadmium and iron concentrations were determined using specific analytical standards, namely cadmium nitrate [Cd(NO_3_)_2_] and ferric chloride hexahydrate [FeCl_3_·6H_2_O].

### 4.12. Fungal Colonization

At the end of the experiment, the plants were removed from the culture medium along with their roots. Using water and a net, all the tiny roots were collected, then washed with distilled water; finally, the water was removed with a paper towel. Then, root samples were fixed in acetic acid-alcohol-formalin solution and placed in tubes containing potassium hydroxide solution (10%) at room temperature for 24 h. In the next step, the roots were kept in a fresh alkaline oxygenated water solution for 30 min and then incubated in an acid solution. Next, they were stained in a color solution containing 0.05% trypan blue and transferred to a color solution with a volume ratio of 1:1:1 of lactic acid/glycerin/water and examined under a light microscope. Finally, 30 pieces of colored roots were randomly placed on a Petri dish (1 cm × 1 cm), the number of colonized cross lines was determined by binocular counting, and the symbiosis percentage was determined [88].

### 4.13. Microbial Population Measurement

To measure the microbial population, 10 g of rhizosphere soil was added to 90 mL of sterile distilled water. Then, using the obtained suspension, a dilution series was prepared up to 8–10, and microbial cultivation was performed using 0.1 mL of the dilution series in sterilized nutrient agar medium in three replicates. Finally, the microbial population was determined by plate-counting bacterial colonies [89].

### 4.14. Scanning Electron Microscope (SEM)

Fresh and healthy plant leaves were prepared in sizes of 4 to 5 mm. Leaf samples were placed in a 3% glutaraldehyde solution at 4 °C for 3 h. Then, washing was performed with distilled water; after that, the samples were dehydrated using methanol and dried [90]. Last, the samples were covered with a layer of gold–palladium, and imaging was performed with a Philips Leo 1450 VL scanning electron microscope (SEM) in the Department of Chemistry, University of Mohaghegh Ardabili, Ardabil, Iran.

### 4.15. Statistical Analysis

The data from biometric, RWC, EL, biochemical, and ion analyses were collected at the end of the experiment. Data were previously assessed for normal distribution (Shapiro–Wilk test) and equal distribution of variance (Bartlett’s test), then analyzed using two-way ANOVA analyses to determine the statistical significance of each set of variables (Cd concentration, mycorrhization), as well as their interaction. ANOVA analyses were followed by a Tukey post hoc test (*p* ≤ 0.05) using the control sample as the reference within each cultivar. Osmolyte data (glycine, betaine, proline) were also analyzed through linear regression analyses. All the computations were performed using GraphPad Prism version 10.2.0 (GraphPad Software, San Diego, CA, USA).

## 5. Conclusions

This work sheds light on the role of beneficial fungi in counteracting the stress effect in Chinese cabbage. In this context, it is worth noting that inoculating Chinese cabbage under Cd stress with *S. indica* is an economical and environmentally friendly solution that protects the plant against stress effects, leading to improved plant growth and yield. These could be achieved by improving plant–water relations and nutrient absorption, the production of secondary metabolites, and the enhancement of both enzymatic and non-enzymatic antioxidants, ultimately leading to a reduction in oxidative stress. Although *S. indica* inoculation had positive effects in this regard, further studies are needed, especially those that include different species, to achieve greater accuracy. Genetic and molecular investigations, as well as biochemical pathways, could be explored to decipher the mechanisms behind the effect of mutualistic relationships in plants.

## Figures and Tables

**Figure 1 plants-14-02773-f001:**
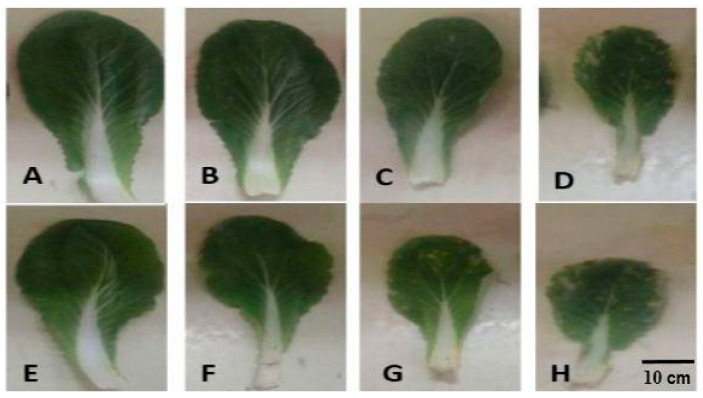
Chinese cabbage leaves under cadmium stress treatments and inoculation with *S. indica* fungus. (**A**–**D**) Plants inoculated under different stress conditions (0, 1, 3, and 4 mM Cd). (**E**–**H**) Non-inoculated (control) plants under cadmium stress conditions (0, 1, 3, and 4 mM).

**Figure 2 plants-14-02773-f002:**
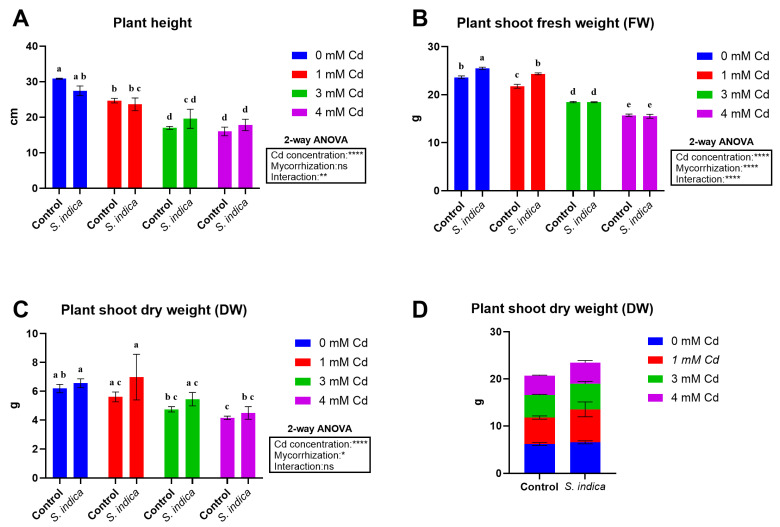
Biometric data related to plant height (**A**), plant shoot fresh weight (**B**), and plant shoot dry weight (**C**,**D**) of Chinese plants inoculated or non-inoculated (control) with *S. indica* fungus and subjected to different stress conditions (0, 1, 3, and 4 mM of cadmium). Values are reported as median values ± SD (n = 3). Different lower-case letters indicate significant differences among samples, as determined by Tukey’s HSD post hoc test following two-way ANOVA for the stress variable (Cd concentration). Asterisks indicate the statistical significance of each factor (Cd concentration, mycorrhization) and their interaction as a result of two-way ANOVA analysis. **** < 0.0001, ** < 0.01, * < 0.05.

**Figure 3 plants-14-02773-f003:**
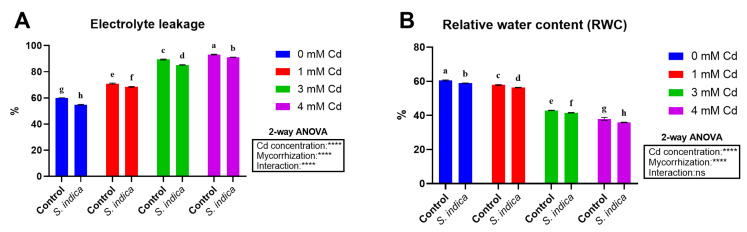
Electrolyte leakage (**A**) and relative water content (**B**) data of Chinese plants inoculated or non-inoculated (control) with *S. indica* fungus and subjected to different stress conditions (0, 1, 3, and 4 mM of cadmium). Values are reported as median values ± SD (n = 3). Different lower-case letters indicate significant differences among samples, as determined by Tukey’s HSD post hoc test following two-way ANOVA for the stress variable (Cd concentration). Asterisks indicate the statistical significance of each factor (Cd concentration, mycorrhization) and their interaction as a result of two-way ANOVA analysis.**** < 0.0001.

**Figure 4 plants-14-02773-f004:**
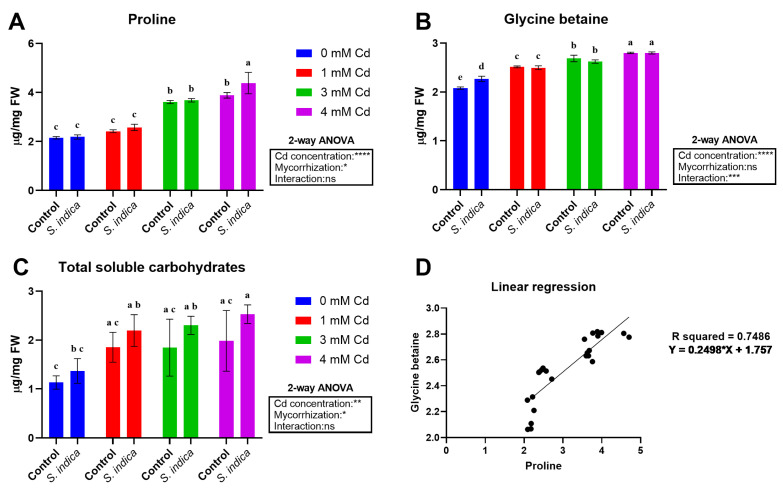
Proline (**A**), glycine betaine (**B**), and total soluble carbohydrate (**C**) data of Chinese plants inoculated or non-inoculated (control) with *S. indica* fungus and subjected to different stress conditions (0, 1, 3, and 4 mM of cadmium). Proline and glycine betaine levels were also analyzed using linear regression analysis (**D**). Values are reported as median values ± SD (n = 3). Different lower-case letters indicate significant differences among samples, as determined by Tukey’s HSD post hoc test following two-way ANOVA for the stress variable (Cd concentration). Asterisks indicate the statistical significance of each factor (Cd concentration, mycorrhization) and their interaction as a result of two-way ANOVA analysis. **** < 0.0001, *** < 0.001, ** < 0.01, * < 0.05.

**Figure 5 plants-14-02773-f005:**
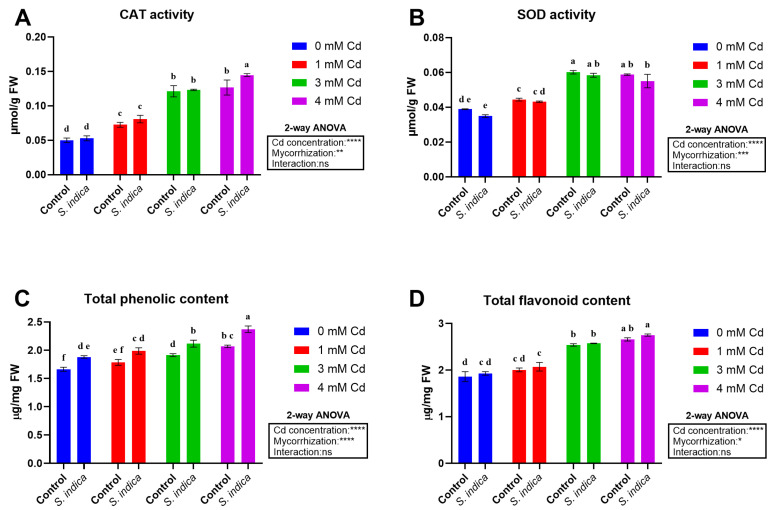
Data related to Catalase (**A**) and Superoxide dismutase (**B**) activities and total phenolic (**C**) and total flavonoid (**D**) contents from Chinese plants inoculated or non-inoculated (control) with *S. indica* fungus and subjected to different stress conditions (0, 1, 3, and 4 mM of cadmium). Proline and glycine betaine levels were also analyzed using linear regression analysis (**D**). Values are reported as median values ± SD (n = 3). Different lower-case letters indicate significant differences among samples, as determined by Tukey’s HSD post hoc test following two-way ANOVA for the stress variable (Cd concentration). Asterisks indicate the statistical significance of each factor (Cd concentration, mycorrhization) and their interaction as a result of two-way ANOVA analysis. **** < 0.0001, *** < 0.001, ** < 0.01, * < 0.05.

**Figure 6 plants-14-02773-f006:**
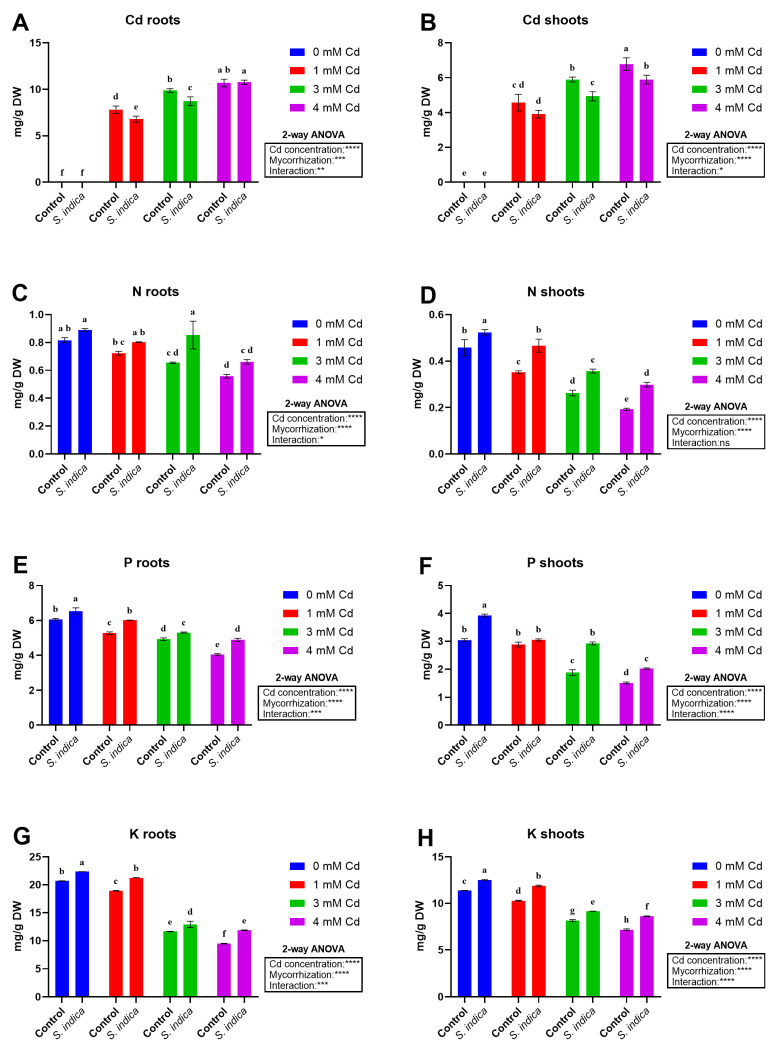
Cadmium (Cd), nitrogen (N), phosphate (P), and potassium (K) concentrations measured in different plant tissues (shoots, roots) in Chinese plants inoculated or non-inoculated (control) with *S. indica* fungus and subjected to different stress conditions (0, 1, 3, and 4 mM of cadmium). (**A**,**C**,**E**,**G**) Root data, (**B**,**D**,**F**,**H**) shoot data. Values are reported as median values ± SD (n = 3). Different lower-case letters indicate significant differences among samples, as determined by Tukey’s HSD post hoc test following two-way ANOVA for the stress variable (Cd concentration). Asterisks indicate the statistical significance of each factor (Cd concentration, mycorrhization) and their interaction as a result of two-way ANOVA analysis. **** < 0.0001, *** < 0.001, ** < 0.01, * < 0.05.

**Figure 7 plants-14-02773-f007:**
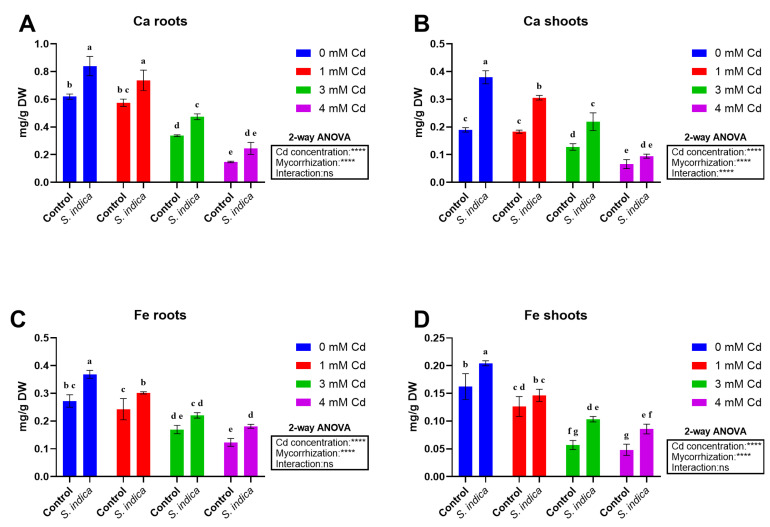
Calcium (Ca) and iron (Fe) concentrations measured in different plant tissues (shoots, roots) in Chinese plants inoculated or non-inoculated (control) with *S. indica* fungus and subjected to different stress conditions (0, 1, 3, and 4 mM of cadmium). (**A**,**C**) Root data, (**B**,**D**) shoot data. Values are reported as median values ± SD (n = 3). Different lower-case letters indicate significant differences among samples, as determined by Tukey’s HSD post hoc test following two-way ANOVA for the stress variable (Cd concentration). Asterisks indicate the statistical significance of each factor (Cd concentration, mycorrhization) and their interaction as a result of two-way ANOVA analysis. **** < 0.0001.

**Figure 8 plants-14-02773-f008:**
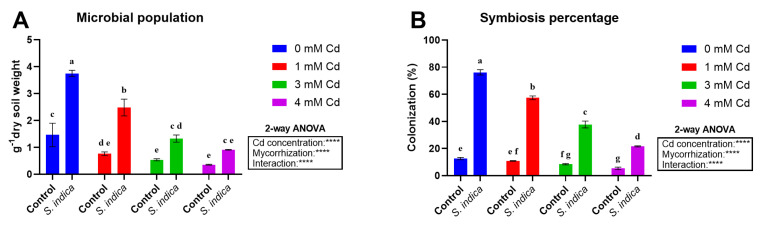
Microbial population (**A**) and symbiosis percentage (**B**) in Chinese plants inoculated or non-inoculated (control) with *S. indica* fungus and subjected to different stress conditions (0, 1, 3, and 4 mM of cadmium). Values are reported as median values ± SD (n = 3). Different lower-case letters indicate significant differences among samples, as determined by Tukey’s HSD post hoc test following two-way ANOVA for the stress variable (Cd concentration). Asterisks indicate the statistical significance of each factor (Cd concentration, mycorrhization) and their interaction as a result of two-way ANOVA analysis. **** < 0.0001.

**Figure 9 plants-14-02773-f009:**
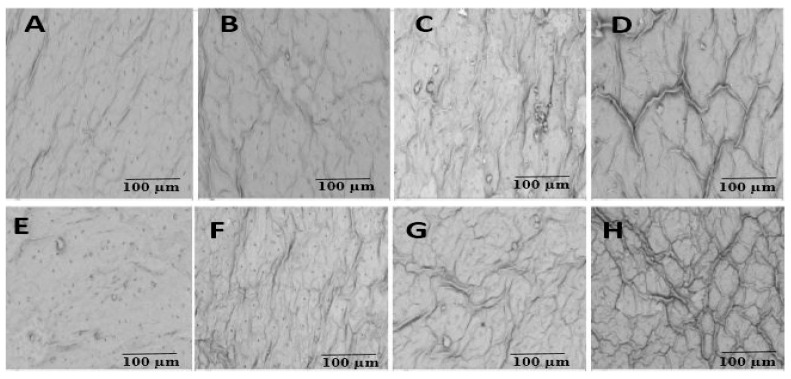
Scanning Electron Microscope (SEM) observations of Chinese cabbage plant leaf stomata with a magnification of 100 µm under cadmium stress conditions and inoculation with *S. indica* fungus. Images (**A**–**D**) include plants inoculated with fungus and cadmium at four levels (0, 1, 3, and 4 mM, respectively). Images (**E**–**H**) include plants without inoculation with fungus and cadmium at four levels (0, 1, 3, and 4 mM, respectively).

**Figure 10 plants-14-02773-f010:**
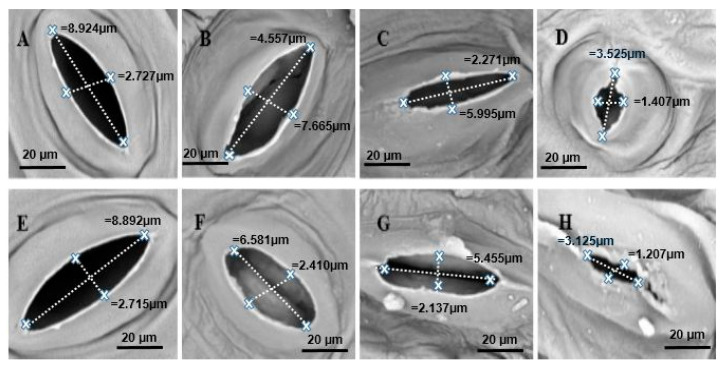
Scanning Electron Microscope (SEM) observations of the stomatal structures with 20 µm magnification under cadmium stress conditions and inoculation with *S. indica* fungus. Images (**A**–**D**) include plants inoculated with fungus and cadmium at four levels (0, 1, 3, and 4 mM, respectively). Images (**E**–**H**) include plants without inoculation with fungus and cadmium at four levels (0, 1, 3, and 4 mM, respectively).

## Data Availability

The raw data supporting the conclusions of this article will be made available by the authors on request.

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
