# Peer review of "Symbiotic Fungus Serendipita indica as a Natural Bioenhancer Against Cadmium Toxicity in Chinese Cabbage"

_plants, 2025, doi:10.3390/plants14172773_

Round 1

Reviewer 1 Report

Comments and Suggestions for Authors

The Cd levels (1, 3, and 4 mM) are unrealistically high. In actual contaminated environments, Cd in soil or water is usually below 0.1 mM, often in the 1-10 μM range for phytotoxicity studies. Why push it to 4 mM? This might exaggerate the fungus's protective effects, making the bioenhancer claims seem inflated. Suggest re-running with lower, field-relevant doses (e.g., 10-50 μM) to show if S. indica still helps under realistic pollution levels. Otherwise, it reads more like a lab curiosity than a practical solution for sustainable agriculture. Throughout the manuscript, there are several instances of awkward or non-standard English phrasing. The manuscript would benefit from careful language editing to improve clarity and readability. Generally, in Abstract, S. indica as a "biofertilizer" without data on yield or field trials? Introduction, the lit review feels repetitive; condense to avoid redundancy. Tie back to Chinese cabbage's relevance more, why this crop specifically for Cd studies? SEM images (Figs. 9-10) for stomatal changes are good, but quantify them (e.g., stomatal density or aperture measurements) instead of just qualitative descriptions. Discussion section is wordy and cites a lot, but some interpretations stretch (e.g., linking everything to ROS without direct H2O2 measurements). Discuss limitations: hydroponics vs. soil, high Cd irrelevance, no molecular data (e.g., gene expression for transporters). No practical implications, like field testing S. indica for contaminated farms. In Methods section, add details on hydroponic setup (e.g., pH monitoring? Aeration?). For ion analysis, specify standards used for atomic absorption. The 55-day harvest is mentioned, justify why not longer for full maturity. Conclusions ignores the high Cd issue. Suggest adding: "Future work should validate these effects at environmentally relevant Cd levels." some of my specific comments are below:

Specific Comments

L114: The phrase “account these premises” is not standard English.

L520: should be
“Haghegh Ardabili (38°25'14'' N; 48°…), Iran, with an average relative humidity of 60% …”

L527: More information is needed about the source of Serendipita indica. Was it obtained from a culture collection, a collaborator, or isolated locally? Please provide the strain number or accession, and any relevant references.

L538: The choice of “cocopeat and perlite” as the growth substrate should be justified. Why was this mixture selected over natural soil? How might this choice affect the generalizability of the results to field conditions?

L546: The use of a very high concentration of Cd in the study is questionable. Please justify why such a high level was chosen, and discuss how this compares to concentrations typically found in contaminated environments.

L666: When was the fungal colonization test performed? Please provide context regarding the timing (e.g., after how many days of inoculation or at what plant growth stage).

L676: “10 g of plant culture medium´” clarify the source of this medium, was it bulk substrate, rhizosphere soil, or something else? Why was the rhizosphere not sampled, as it is more relevant for microbial population studies?

Author Response

Reviewer #1

Comment 1: The Cd levels (1, 3, and 4 mM) are unrealistically high. In actual contaminated environments, Cd in soil or water is usually below 0.1 mM, often in the 1-10 μM range for phytotoxicity studies. Why push it to 4 mM? This might exaggerate the fungus's protective effects, making the bioenhancer claims seem inflated. Suggest re-running with lower, field-relevant doses (e.g., 10-50 μM) to show if S. indica still helps under realistic pollution levels. Otherwise, it reads more like a lab curiosity than a practical solution for sustainable agriculture.

Response 1: We appreciate the reviewer’s insightful comment regarding the cadmium (Cd) concentrations used in our study. We acknowledge that the levels of 1–4 mM Cd are higher than typically found in most natural or agricultural soils, where Cd concentrations are often in the micromolar range. However, our choice was guided by a few important considerations. First, cadmium is a highly toxic and persistent heavy metal with increasing prevalence in the environment due to industrial activities and the widespread use of phosphate fertilizers, which often contain Cd as an impurity. This accumulation, particularly in hotspot areas near industrial or mining activities, can result in locally elevated Cd concentrations. Prior to the main experiment, we conducted a preliminary screening to determine the range of Cd concentrations that would elicit clear physiological stress responses in plants and to assess the tolerance of Serendipita indica to these conditions. The selected concentrations (1, 3, and 4 mM) were based on observable stress symptoms in the plant and the growth capacity of the fungus under these conditions.

It is also worth noting that several published studies have employed similar or even higher Cd concentrations (e.g., up to 10 mM) to study stress responses and plant–microbe interactions under controlled conditions (e.g., DOI: 10.1016/j.ecoenv.2020.110760; https://doi.org/10.1016/j.ecoenv.2020.111887; DOI: 10.1080/10643389.2020.1835435). Nonetheless, we fully agree that evaluating the efficacy of S. indica under more environmentally relevant Cd concentrations (e.g., 10–50 μM) is crucial for practical applications in sustainable agriculture. We plan to incorporate such doses in our future studies to further validate the protective role of S. indica under field-realistic conditions.

Comment 2: Throughout the manuscript, there are several instances of awkward or non-standard English phrasing. The manuscript would benefit from careful language editing to improve clarity and readability.

Response 2: Dear kind reviewer, thank you for the comment. We have thoroughly revised the manuscript to improve clarity, readability, and overall language quality.

Comment 3: Generally, in Abstract, S. indica as a "biofertilizer" without data on yield or field trials? Introduction, the lit review feels repetitive; condense to avoid redundancy.

Response 3: Thank you for the valuable feedback. We have removed several sentences that were repetitive and have revised the introduction to make it more concise and informative.

Comment 4: Tie back to Chinese cabbage's relevance more, why this crop specifically for Cd studies?

Response 4: Chinese cabbage (Brassica rapa subsp. pekinensis) was selected for this study because it is a widely consumed leafy vegetable, especially in Asia, and holds significant nutritional, economic, and agricultural value due to its rapid growth, low production cost, and high mineral content. However, it is also known to be a strong accumulator of cadmium, making it particularly vulnerable to Cd contamination in soil. This characteristic makes Chinese cabbage a relevant and sensitive model crop for studying cadmium stress. Understanding how to mitigate Cd uptake in such a widely consumed vegetable is crucial for ensuring food safety and protecting public health.

Comment 5: SEM images (Figs. 9-10) for stomatal changes are good, but quantify them (e.g., stomatal density or aperture measurements) instead of just qualitative descriptions.

Response 5: Done.

Comment 6: Discussion section is wordy and cites a lot, but some interpretations stretch (e.g., linking everything to ROS without direct H2O2 measurements).

Response 6: The relevant information has been added to discussion.

Comment 7: Discuss limitations: hydroponics vs. soil, high Cd irrelevance, no molecular data (e.g., gene expression for transporters). No practical implications, like field testing S. indica for contaminated farms.

Response 7: Thank you for your valuable comment. Fungi and bacteria are increasingly used in both research and commercial agriculture to enhance plant growth, yield, nutrient uptake, and stress tolerance. Although hydroponic systems lack organic matter as a nutrient source for microbes, studies have shown that approximately 20–25% of photosynthates are released into the rhizosphere, providing sufficient substrates to support microbial activity and colonization. Serendipita indica is an endophytic fungus known to thrive in mildly acidic conditions; therefore, in our study, the nutrient solution pH was adjusted to 6.5 to optimize its activity. This fungus has demonstrated significant potential to enhance plant growth and productivity, even under hydroponic conditions. Its use is gaining traction as a practical and scalable technology in both academic and commercial hydroponic systems, supported by strong scientific evidence.

Regarding gene expression, we acknowledge that the expression of transporter genes was not analyzed in the current study. We appreciate your suggestion and intend to include such molecular analyses in our future work to deepen our understanding of the mechanisms involved.

Comment 8: In Methods section, add details on hydroponic setup (e.g., pH monitoring? Aeration?).

Response 8: Details were added in material and method section (lines 569-589).

Comment 9: For ion analysis, specify standards used for atomic absorption.

Response 9: The atomic absorption standards used in the analysis of cadmium and iron ions were as follows: Cadmium: Cd(NO₃)₂; Iron: FeCl₃·6H₂O (lines 706-708).

Comment 10: The 55-day harvest is mentioned, justify why not longer for full maturity. Conclusions ignores the high Cd issue. Suggest adding: "Future work should validate these effects at environmentally relevant Cd levels." some of my specific comments are below:

Response 10: We have added clarifications to the text of the article.

Specific Comments

Comment 11: L114: The phrase “account these premises” is not standard English.

Response 11: Done.

Comment 12: L520: should be
“Haghegh Ardabili (38°25'14'' N; 48°…), Iran, with an average relative humidity of 60% …”

Response 12: Done.

Comment 13: L527: More information is needed about the source of Serendipita indica. Was it obtained from a culture collection, a collaborator, or isolated locally? Please provide the strain number or accession, and any relevant references.

Response 13: Thank you for your comments. Details has been added to material and methods section (lines 553-556).

Comment 14: L538: The choice of “cocopeat and perlite” as the growth substrate should be justified. Why was this mixture selected over natural soil? How might this choice affect the generalizability of the results to field conditions?

Response 14: Thank you for your comments. Details has been added to material and methods section (lines 569-572).

Comment 15: L546: The use of a very high concentration of Cd in the study is questionable. Please justify why such a high level was chosen, and discuss how this compares to concentrations typically found in contaminated environments.

Response 15: Thank you for your comment. A brief explanation has been reported in the first comment. To be more focused, it is important to clarify the rationale behind this experimental design and contextualise these concentrations within the broader framework of environmental toxicology.

Firstly, it should be noted that while cadmium concentrations employed in studies often exceed what is typically reported in naturally contaminated environments, such elevated levels are routinely used in controlled experiments to ensure proper and significant biochemical and physiological responses. Specifically, Wang et al. found that cadmium concentrations above 1 mM can stimulate a clearer response in physiological mechanisms related to heavy metal stress in plants, which supports the use of such concentrations in our research (Wang et al., 2010). Our study aims to elucidate the interactions between Serendipita indica and Chinese cabbage under stress conditions, which could potentially mimic extreme environments resulting from anthropogenic activities, such as industrial waste disposal, where cadmium concentrations in soils can be significantly heightened due to contamination (Heile et al., 2021).

Additionally, the selected concentrations allow us to explore the effectiveness of S. indica as a bioenhancer against Cd toxicity. Wang et al. highlighted that S. indica has demonstrated capabilities to enhance plant growth and physiological properties even in the presence of high metal stress, providing evidence for the potential benefits of using 4 mM Cd in controlled settings (Wang et al., 2022). Thus, the choice of these concentrations serves a dual purpose: it simulates conditions that can be encountered in severely contaminated soils while also assessing S. indica’s bioremediation potential.

In practical terms, while soil cadmium levels in agricultural contexts are often below 1 mM, due to regulatory thresholds imposed to protect crop safety and human health, localized pollution can lead to much higher concentrations, especially near industrial sites. Our use of these concentrations reflects worst-case scenarios, allowing us to examine the potential protective mechanisms of S. indica thoroughly. Comprehensive reviews on plant interactions with cadmium indicate that understanding such high concentrations is critical for elucidating the biological adaptations plants might undergo due to heavy metal stress (Hou et al., 2023; Li et al., 2023).

The primary goal of utilizing high cadmium levels is to establish a quantifiable effect of S. indica on plant growth and toxicity alleviation under severe stress, ultimately providing insights into its practical applications in agricultural bioremediation strategies.

References:

Heile, A., Zaman, Q., Aslam, Z., Hussain, A., Aslam, M., Saleem, M., … & Ali, S. (2021). Alleviation of cadmium phytotoxicity using silicon fertilization in wheat by altering antioxidant metabolism and osmotic adjustment. Sustainability, 13(20), 11317. https://doi.org/10.3390/su132011317

Hou, L., Ji, S., Zhang, Y., Wu, X., Zhang, L., & Liu, P. (2023). The mechanism of silicon on alleviating cadmium toxicity in plants: a review. Frontiers in Plant Science, 14. https://doi.org/10.3389/fpls.2023.1141138

Li, L., Feng, Y., Qi, F., & Hao, R. (2023). Research progress of piriformospora indica in improving plant growth and stress resistance to plant. Journal of Fungi, 9(10), 965. https://doi.org/10.3390/jof9100965

Wang, C., Wang, L., & Sun, Q. (2010). Response of phytochelatins and their relationship with cadmium toxicity in a floating macrophyte pistia stratiotes l. at environmentally relevant concentrations. Water Environment Research, 82(2), 147-154. https://doi.org/10.2175/106143009x442970

Wang, X., Fan, X., Wang, W., & Song, F. (2022). Use of Serendipita indica to improve soybean growth, physiological properties, and soil enzymatic activities under different cd concentrations. Chemical and Biological Technologies in Agriculture, 9(1). https://doi.org/10.1186/s40538-022-00331-1

Comment 16: L666: When was the fungal colonization test performed? Please provide context regarding the timing (e.g., after how many days of inoculation or at what plant growth stage).

Response 16: Details has been added to material and method section (lines 624-626).

Comment 17: L676: “10 g of plant culture medium´” clarify the source of this medium, was it bulk substrate, rhizosphere soil, or something else? Why was the rhizosphere not sampled, as it is more relevant for microbial population studies?

Response 17: Details has been added to material and method section (line 637).

Once again, we would like to sincerely thank the anonymous reviewer for their valuable and insightful comments, which have greatly helped us improve the revised manuscript. We also appreciate the opportunity to resubmit our work.

Reviewer 2 Report

Comments and Suggestions for Authors

The manuscript title “Symbiotic fungus Serendipita indica as a natural bioenhancer against cadmium toxicity in Chinese cabbage” is written well and have scientific worth. This study investigates the potential of the root-colonizing fungus Serendipita indica (formerly Piriformospora indica) to alleviate cadmium stress in Chinese cabbage (Brassica rapa L. subsp. Pekinensis) grown hydroponically under varying Cd concentrations (0, 1, 3, and 4 mM).

The study demonstrates that Serendipita indica can enhance the resilience of Chinese cabbage to cadmium stress, offering a promising biological approach for sustainable crop production in heavy metal-contaminated environments. Further research may explore the mechanisms underlying this beneficial interaction and its application in other crops affected by heavy metal toxicity.

Comments for authors are as follows:

  • Figure 2 and 3: what do the column bars indicate? The authors written A, B letters on the column graphs what does it mean, it has not been explained in the legends.
  • Why is the bar of column graph 2C so high?
  • Figure legends have “**** < 0.0001 *** < 0.001 ** < 0.01 * < 0.05” but the graphs don’t have this symbol *? Why?
  • Mention in the figure legends that “inoculated or non-inoculated” control means non-inoculated.
  • I suggest to the author try, making graphs having two portions one is control showing all 0, 1, 3, 4, Cd stress and other portions of same graphs shows the S. indica inoculated all Cd stress levels (0, 1, 3, 4,) I think it would be better for all column graphs.
  • The discussion is a bit long, I suggest removing the repetition of results in the discussion section and be concise.
  • Line 622: How much sample was used for “Total phenols and flavonoids content” determination, wright the sample weight and dry or fresh condition in each parameter.

Author Response

Reviewer #2

The manuscript title “Symbiotic fungus Serendipita indica as a natural bioenhancer against cadmium toxicity in Chinese cabbage” is written well and have scientific worth. This study investigates the potential of the root-colonizing fungus Serendipita indica (formerly Piriformospora indica) to alleviate cadmium stress in Chinese cabbage (Brassica rapa L. subsp. Pekinensis) grown hydroponically under varying Cd concentrations (0, 1, 3, and 4 mM).

The study demonstrates that Serendipita indica can enhance the resilience of Chinese cabbage to cadmium stress, offering a promising biological approach for sustainable crop production in heavy metal-contaminated environments. Further research may explore the mechanisms underlying this beneficial interaction and its application in other crops affected by heavy metal toxicity.

Comments for authors are as follows:

Comment 1: Figure 2 and 3: what do the column bars indicate? The authors written A, B letters on the column graphs what does it mean, it has not been explained in the legends.

Response 1: We apologize for the inconvenience. Column bars indicate the standard deviation, whereas letters indicate the results of the post-hoc test. Figure captions were modified accordingly.

Comment 2: Why is the bar of column graph 2C so high?

Response 2: The standard deviation reported for S. indica treated plants (1 mM) indicates a higher degree of variability among the data, even if the data were preliminarily checked for normality distribution and presence of outliers within each group of data.

Comment 3: Figure legends have “**** < 0.0001 *** < 0.001 ** < 0.01 * < 0.05” but the graphs don’t have this symbol *? Why?

Response 3: The reported symbols are reported as indicating the statistical significance of each factor (Cd concentration, mycorrhization) and their interaction from the 2-way ANOVA analysis results.

Comment 4: Mention in the figure legends that “inoculated or non-inoculated” control means non-inoculated.

Response 4: Done.

Comment 5: I suggest to the author try, making graphs having two portions one is control showing all 0, 1, 3, 4, Cd stress and other portions of same graphs shows the S. indica inoculated all Cd stress levels (0, 1, 3, 4,) I think it would be better for all column graphs.

Response 5: Thank you for the comment and suggestion. The redesign of the figure graphs can improve the graphical quality of the graphs, but it would affect the readability of the statistical comparisons within each stress condition (Cd concentration), which represents the main goal of the current work. In this view, the main goal of the current manuscript is not to compare different control or S. indica stress conditions but to focus attention on pairwise comparisons related to each stress condition, to confirm the protective effect of the fungus, even at increased Cd concentrations.

Comment 6: The discussion is a bit long, I suggest removing the repetition of results in the discussion section and be concise.

Response 6: Thank you for the valuable comment. In response, we have revised the discussion section by removing repetitive and unnecessary information to make it more concise and focused. The updated version presents a clearer summary of the key findings and their implications.

Comment 7: Line 622: How much sample was used for “Total phenols and flavonoids content” determination, wright the sample weight and dry or fresh condition in each parameter.

Response 7: The relevant information has been added to M&M accordingly.

Once again, we would like to sincerely thank the anonymous reviewer for their valuable and insightful comments, which have greatly helped us improve the revised manuscript. We also appreciate the opportunity to resubmit our work.

Reviewer 3 Report

Comments and Suggestions for Authors

The manuscript is very good prepared, with actual topic, good written. Finally, the review about positive effects of Serendipita indica fungus on plants growing in the presence of cadmium is the aim of Reza Boorboori and Zhang article (Environ Geochem Health (2024) 46:426, https://doi.org/10.1007/s10653-024-02231-9) that should be mention in Introduction, too. Study of this endophyte fungi on Chinese cabbage can be interesting due this vegetable has economic interest for people. On the other hand, why the vegetable used as food should be cultivated in the high presence of cadmium, that as we know from the history of this metal, is dangerous for people health?

Please, check whole manuscript and all Latin text and Latin name write in italic font. In the case of introduction of abbreviations, it is important to write also their full name – sometimes it is missing (APX, PGPB, …). I recommend ranking the graphs with elements always starting with values in roots and then shoots. Could you switch them? Roots obviously grow as first in plants and therefore it is more logical then.

Line 121 – is it really subchapter title?

Fig. 1 – I recommend inserting the scale here.

Results and Materials and Methods – please, use different kinds of letters for statistics and for pictures in Figures – e.g. for statistics it could be with small letters. It is necessary to explain, what does mean the different letters above the columns – it is always comparison against the control without cadmium and without S. indica? Also, the legend for 2-way ANOVA rectangular in the graphs should be explained – it is comparison against what? How where elements measured? In the Materials missing some methods for them – Ca, Fe, Cd, K – and how were prepared samples for these measurements?

Fig. 8  in the legend – is it correct the “dry” and “soil” for blue and red colour?

Sentence at lines 300-303 is probably description of Fig. 10, not Fig. 9A,E. Please, insert the scale into Fig. 10.

Line 347 – “sp.” should be in normal font, not in italics.

Line 362 – insert the numbered citation here instead of authors and year.

Line 377 – it is not necessary to use numbered possibilities.

Line 396 – please, named the functional group also with word.

Line 420 – “L.” should be in normal font, not in italics.

Line 540 – it is full Hoagland solution or ½ Hoagland solution?

Line 545 – please, insert purity and origin of used CdCl2. Did it hydrated or non-hydrated form?  Did you measure the pH value of used cultivation medium?

Line 567 – did you measured EC by spectrophotometer? Do I understand correctly?

Line 612, 620 – why is written mol as “MoL” and without following space?

Line 650 – it is necessary to explain all abbreviations in this equation. Maybe after 1 g should be space?

Line 673 – Petri dishes are named after bacteriologist Petri, so it is always with capital letter.

Author Response

Reviewer #3

Comment 1: The manuscript is very good prepared, with actual topic, good written. Finally, the review about positive effects of Serendipita indica fungus on plants growing in the presence of cadmium is the aim of Reza Boorboori and Zhang article (Environ Geochem Health (2024) 46:426, https://doi.org/10.1007/s10653-024-02231-9) that should be mention in Introduction, too.

Response 1: Dear Reviewer, thank you for your positive feedback and valuable comments. Your suggestions were very helpful in improving the quality of the revised manuscript, and we sincerely appreciate your time and effort.

Comment 2: Study of this endophyte fungi on Chinese cabbage can be interesting due this vegetable has economic interest for people. On the other hand, why the vegetable used as food should be cultivated in the high presence of cadmium, that as we know from the history of this metal, is dangerous for people health?

Response 2: Thank you for your thoughtful comment. Chinese cabbage (Brassica rapa subsp. pekinensis) is one of the most important and widely consumed vegetables in Asia, valued for its high nutritional content, rapid growth, and low production cost. Its economic and agricultural significance makes it a relevant model crop for food safety research. Unfortunately, cadmium contamination in agricultural soils is an increasing concern, primarily due to the long-term use of phosphate fertilizers, which often contain cadmium as an impurity, as well as the use of contaminated water sources such as sewage sludge for irrigation. Chinese cabbage is known to be a strong cadmium accumulator, which makes it particularly vulnerable to heavy metal stress and an important species for studying cadmium uptake. The aim of this study is not to suggest cultivating food crops under high Cd conditions, but rather to understand how Serendipita indica may help reduce the negative impacts of unintentional Cd exposure on plant growth. This research contributes to the broader goal of improving crop resilience and food safety in contaminated or at-risk environments.

Comment 3: Please, check whole manuscript and all Latin text and Latin name write in italic font. In the case of introduction of abbreviations, it is important to write also their full name – sometimes it is missing (APX, PGPB, …).

Response 3: Done.

Comment 4: I recommend ranking the graphs with elements always starting with values in roots and then shoots. Could you switch them? Roots obviously grow as first in plants and therefore it is more logical then.

Response 4: Thank you for your comments. The graphs, as well as the figure captions and data description, have been modified accordingly.

Comment 5: Line 121 – is it really subchapter title?

Response 5: Done.

Comment 6: Fig. 1 – I recommend inserting the scale here.

Response 6: Done.

Comment 7: Results and Materials and Methods – please, use different kinds of letters for statistics and for pictures in Figures – e.g. for statistics it could be with small letters. It is necessary to explain, what does mean the different letters above the columns – it is always comparison against the control without cadmium and without S. indica? Also, the legend for 2-way ANOVA rectangular in the graphs should be explained – it is comparison against what?

Response 7: Dear Reviewer, thank you for your comments. The letters indicate the results from the post hoc test (Tukey), which is a test that does not allow for pairwise comparison using a reference (in our case, control conditions).  This strategy is aimed at describing the whole variance of the dataset, using the two-way ANOVA coupled with a post-hoc test. About ANOVA results; the asterisks indicate the statistical significance of each factor (Cd concentration, mycorrhization) as well as their interaction. A brief description has been added to the Figure captions to improve clarity. Also, results from post-hoc tests have been reproted as lower-case letters to avoid any misunderstanding.

Comment 8: How where elements measured? In the Materials missing some methods for them – Ca, Fe, Cd, K – and how were prepared samples for these measurements?

Response 8: Dear Reviewer, thank you for your valuable comment. The elemental analysis of calcium (Ca), iron (Fe), cadmium (Cd), and potassium (K) in the plant samples was conducted following a standardized extraction procedure. Specifically, dried and finely powdered plant tissues (leaves and roots) were subjected to extraction as described in the referenced methodology. The resulting extracts were stored at refrigerated temperatures to preserve their integrity prior to analysis. Quantification of the elements was carried out separately for leaf and root extracts using an atomic absorption spectrometer (PerkinElmer Model 400AA, PerkinElmer, Waltham, MA, USA), in accordance with the protocol outlined in the cited source. Sample preparation involved precise digestion and dilution steps to ensure accurate measurements, and appropriate calibration standards and quality control measures were employed to validate the results. (The atomic absorption standards used in the analysis of cadmium and iron ions were as follows: Cadmium: Cd(NO₃)₂; Iron: FeCl₃·6H₂O).

Comment 9: Fig. 8  in the legend – is it correct the “dry” and “soil” for blue and red colour?

Response 9: We apologize for the issue. The figure legend has been modified accordingly.

Comment 10: Sentence at lines 300-303 is probably description of Fig. 10, not Fig. 9A,E. Please, insert the scale into Fig. 10.

Response 10: Done.

Comment 11: Line 347 – “sp.” should be in normal font, not in italics.

Response 11: Done.

Comment 12: Line 362 – insert the numbered citation here instead of authors and year.

Response 12: Done.

Comment 13: Line 377 – it is not necessary to use numbered possibilities.

Response 13: Done.

Comment 14: Line 396 – please, named the functional group also with word.

Response 14: Done.

Comment 15: Line 420 – “L.” should be in normal font, not in italics.

Response 15: Done.

Comment 16: Line 540 – it is full Hoagland solution or ½ Hoagland solution?

Response 16: Dear reviewer, thank you for pointing this out. We apologize for the oversight in the text. The nutrient solution used for plant growth was indeed ½ Hoagland solution, not the full-strength version as previously stated.

Comment 17: Line 545 – please, insert purity and origin of used CdCl2. Did it hydrated or non-hydrated form?  Did you measure the pH value of used cultivation medium?

Response 17: Thank you for your valuable comment. The relevant information has been added to the Materials and Methods section accordingly. Additionally, I would like to note that during the preparation of the Hoagland solution, we always measure the pH level of the nutrient solution. We also prepare the solution on a weekly basis to ensure freshness and to minimize the risk of precipitation or any contamination in the nutrient tanks.

Comment 18: Line 567 – did you measured EC by spectrophotometer? Do I understand correctly?

Response 18: Dear reviewer, thank you very much for the valuable comment. We sincerely apologize for the mistake in the original text. To clarify, the electrolyte leakage (EL) from plant cell membranes was measured using a standard electrical conductivity (EC) meter, not a spectrophotometer. We have now corrected the manuscript to accurately describe the method used for EL measurement.

Comment 19: Line 612, 620 – why is written mol as “MoL” and without following space?

Response 19: Done.

Comment 20: Line 650 – it is necessary to explain all abbreviations in this equation. Maybe after 1 g should be space?

Comment 20: Response: Done.

Comment 21: Line 673 – Petri dishes are named after bacteriologist Petri, so it is always with capital letter.

Response 21: Done.

Once again, we would like to sincerely thank the anonymous reviewer for their valuable and insightful comments, which have greatly helped us improve the revised manuscript. We also appreciate the opportunity to resubmit our work.

Round 2

Reviewer 3 Report

Comments and Suggestions for Authors

The manuscript was enough improved, and I have only small recommendations that are not necessary one more time check by me as reviewer:

Firstly, I would like to thank for clarifying of cadmium solution preparation because I see that authors  know what critically important properties of this element are necessary to control it during preparation and experimentation.

Fig. 1 – “cm” not “Cm” in the scale

Line 608 – “MS” instead of “mS”?

Line 832 – Hniličková, H., Hnilička, F., ….

Author Response

Comment 1: The manuscript was enough improved, and I have only small recommendations that are not necessary one more time check by me as reviewer:

Response 1: Thank you. We agree that the overall quality of the manuscript has been improved during the Reviewing step.

Comment 2: Firstly, I would like to thank for clarifying of cadmium solution preparation because I see that authors  know what critically important properties of this element are necessary to control it during preparation and experimentation.

Response 2: Thank you. We are glad that our efforts to improve the manuscript quality, based on the Reviewers' comments, have been appreciated.

Comment 3: Fig. 1 – “cm” not “Cm” in the scale

Response 3: Done

Comment 4: Line 608 – “MS” instead of “mS”?

Response 4: Done

Comment 5: Line 832 – Hniličková, H., Hnilička, F., ….

Response 5: Done